# Are Adenomyosis and Endometriosis Phenotypes of the Same Disease Process?

**DOI:** 10.3390/biom14010032

**Published:** 2023-12-25

**Authors:** Marwan Habiba, Sun-Wei Guo, Giuseppe Benagiano

**Affiliations:** 1Department of Health Sciences, University of Leicester and University Hospitals of Leicester, Leicester LE1 5WW, UK; 2Department of Biochemistry and Molecular Biology, Research Institute, Shanghai Obstetrics and Gynecology Hospital, Fudan University, Shanghai 200011, China; hoxa10@outlook.com; 3Faculty of Medicine and Surgery, Sapienza University of Rome, 00161 Rome, Italy; pinoingeneva@bluewin.ch; 4Geneva Foundation for Medical Education and Research, 1202 Geneva, Switzerland

**Keywords:** adenomyosis, endometriosis, pathogenesis, KRAS, epidemiology

## Abstract

In recent literature reviews, we concluded that the possibility that endometrial molecular aberrations are the sole or a necessary determinant of endometriosis and the Tissue Injury and Repair (TIAR) theory are yet to be convincingly proven. Here, we critically examine the theory that adenomyosis and endometriosis represent different phenotypes of a single disease. A common etiopathology for adenomyosis and endometriosis has been suggested because both conditions entail the presence of endometrial tissue at locations other than the lining of the uterus. There are wide differences in reported disease incidence and prevalence and, consequently, in estimates of the coexistence of both conditions. There are some similarities but also differences in their clinical features and predisposing factors. Each condition has a range of subtypes. These differences alone pose the question of whether subtypes of endometriosis and adenomyosis have different etiopathologies, and, in turn, this raises the question of whether they all share a common etiology. It is debatable whether the recognized differences between the eutopic endometrium in adenomyosis and endometriosis compared to those in unaffected women are the cause or the effect of the disease. The finding of common mutations, particularly of *KRAS*, lend support to the notion of shared predisposing factors, but this alone is insufficient evidence of causation.

## 1. Introduction

As the pathophysiology of endometriosis and of adenomyosis remain largely speculative, much can be gained from a critical analysis of our current understanding. To elucidate this issue, we undertook two reviews of current theories about the origin of the conditions [1,2]. In relation to the so-called “*endometrial determinism*” theory (based on the hypothesis that an abnormal endometrium is the sole and foremost predisposing factor for endometriosis [3]), we concluded that the possibility that endometrial molecular aberrations are the sole or a necessary determinant of endometriosis is yet to be proven [1]. We also examined the Tissue Injury and Repair (TIAR) theory [4,5], which suggests that adenomyosis and endometriosis are two phenotypes of the same disease, and concluded that the theory is yet to be convincingly articulated [2].

The link between both conditions has been suggested in theories that identified endometrial abnormalities in instances where both conditions coexist, particularly in patients with infertility. When—during the second half of the 19th century—the presence of epithelial cells on the peritoneal surface and within the myometrium was first identified, endometriosis (except for ovarian endometriomas) and adenomyosis were considered under the common name ‘*adenomyoma’*. Subsequently, in the 1920s, as they came to be regarded as separate entities, the terms *adenomyosis* and *endometriosis* gained wide acceptance. Interestingly, the term endometriosis also came to be used as an overarching term that encompasses instances where endometrial tissue is present outside the lining of the uterus. Adenomyosis came to be referred to as endometriosis *interna* to distinguish it from cases where the aberrant endometrium is present outside the uterus, which was referred to as endometriosis *externa* (e.g., [6]).

This topic has been addressed in several recent reviews focusing on the hypothesis that extrinsic (or external, or focal) adenomyosis located in the outer myometrium may result from pelvic endometriosis, especially the rectal and bladder deep variants [7,8].

We have previously proposed that dysfunction of the myometrial junctional zone (JZ) and aberrations in the eutopic and heterotopic endometria indicate that adenomyosis and endometriosis may share a common origin [9,10]. This was based on data that indicate that the two diseases have common aberrations and dysfunctions in both the eutopic and the ectopic endometrium. This causes a reaction in the JZ. At the time, we opined that adenomyosis and endometriosis seem to share a common origin from abnormal eutopic endometrium and myometrium [9]. We also presented a summary of observed similarities and differences between the endometrium in adenomyosis and in endometriosis [11].

More recently, one of us questioned whether adenomyosis is simply endometriosis of the uterus, either from the perspective of pathogenesis or pathophysiology, and proposed a new hypothesis implicating an “endometrial-myometrial interface disruption (EMID)” in variants of adenomyosis resulting from iatrogenic trauma to the JZ myometrium, concluding that from a pathogenetic standpoint, adenomyosis is not simply endometriosis of the uterus. Another review addressed a possible pathogenetic role of Schwann cells (a variety of glial cells that keep peripheral nerve fibers alive) in the JZ area resulting from EMID [12].

A very recent review by Bulun et al. [13] focused on existing evidence that the pathophysiology of both conditions is ‘extremely similar’. Briefly, they argued that both tissues originate from the eutopic endometrium and that oligoclones of endometrial glandular epithelial cells carrying somatic mutations and attached stromal cells may either retrogradely travel to peritoneal surfaces or organs, producing endometriosis, or penetrate into the myometrium, giving rise to adenomyosis.

The enhanced epithelial cell survival and growth are enabled by the presence of somatic epithelial mutations and epigenetic abnormalities in stromal cells. Activating mutations of KRAS are the most commonly found genetic variants in endometriotic epithelial cells, whereas the adenomyotic epithelial cells almost exclusively bear KRAS mutations [14].

Theories rooted in the identification of endometrial abnormalities, particularly in women with both endometriosis and adenomyosis, provide support for a link between the two conditions. Whether endometriosis and adenomyosis are two phenotypes of the same disease is an intriguing question. At its core is whether the two conditions have a shared origin or trigger. Notwithstanding this, a common origin does not imply that they are one and the same disease, since they differ in their gross, histological, and clinical manifestations and risk factors [2].

Having carried out critical reviews of the major theories, we felt an obligation to critically review our own work in the hope of advancing the debate on the possibility that adenomyosis and endometriosis represent different phenotypes of a single disease.

## 2. Materials and Methods

### Search Strategy and Inclusion Criteria

This review includes all papers indexed in PubMed until 1 September 2023, and then updated on 13 October. Search items were as follows: (a) adenomyosis plus etiology or pathophysiology (n = 877 articles); (b) endometriosis plus etiology or pathophysiology (n = 8862); (c) combination of (a) plus (b) (n = 496 articles), which were searched manually based on the title and abstract to identify articles that addressed joint pathophysiology; (d) adenomyosis plus incidence or prevalence (n = 436 articles; (e) endometriosis plus incidence or prevalence (n = 2663 articles); and (f) combination of (d) plus (e) (n = 247 articles), which were searched manually based on the title and abstract (Table 1). This yielded a total of 36 articles that were reviewed in full text. Additionally, the reference lists of all papers eligible for review were manually checked for relevant articles.

## 3. Phenotypes of Endometriosis and Adenomyosis

Endometriosis is now more clearly recognized as entailing the presence of endometrial epithelium and stroma at ectopic sites outside the uterus. The most recent definition from the World Health Organization reads, “*Endometriosis is a disease in which tissue similar to the lining of the uterus grows outside the uterus. It can cause severe pain in the pelvis and make it harder to get pregnant*” [15]. However, beyond this seemingly clear definition, developing a unified theory of endometriosis has been elusive because of the diversity of locations and clinical phenotypes. The implications of the condition differ based on a number of factors, such as whether it affects the ovary, the peritoneal surface, is deeply infiltrating, or affects peritoneal (or even extra-peritoneal) organs. Thus, even framing the diverse clinical manifestations of the disease under a unified pathogenetic theory is in itself quite challenging; for this reason, a call has been made for “reclassifying endometriosis as a syndrome.”

However, it has been long argued that the various phenotypes of endometriosis cannot be explained by a single theory. Indeed, whereas for peritoneal and ovarian variants the most widely accepted theory involves a retrograde flux of menstruation, as first proposed by Sampson [16], Sampson himself classified endometriosis into several phenotypes, each having a separate pathogenetic mechanisms [17]:-A variant recognized today as *adenomyosis*, where the ectopic endometrial tissue raises by direct extension into the uterine wall.-A type resulting from retrograde menstruation, namely, the peritoneal and ovarian implantation of endometrial cells and stroma.-The transplantation of the ectopic tissue as a consequence of dissemination due to surgical wounds.-A “*metastatic*” variant as a consequence of lymphatic or hematogenous microembolization of endometrial cells.-A developmentally determined variant where the presence of ectopic endometrium is the consequence of embryonic remnants.

Features of the “metastatic” phenotype have been recently reviewed [18,19] (Table 2).

### 3.1. Phenotypes of Endometriosis

Endometriosis can affect different sites in the peritoneum, the ovary, or the rectovaginal septum (as well as more distant sites, such as the lungs). Histological investigations have suggested that peritoneal endometriosis, ovarian endometrioma and rectovaginal endometriosis are three separate entities with different pathogeneses [20]. There are similarities between proliferative eutopic endometrium and red peritoneal nodules; in more advanced disease, these evolve into black lesions and eventually into plaques of old collagen [21]. Black lesions rarely demonstrate typical progestational changes [20]. Affection of the rectovaginal septum contain aggregates of smooth muscle and may represent a form of deep-infiltrating endometriosis, now renamed deep endometriosis (DE) [22], or adenomyotic nodules originating from Müllerian rests [20].

In conclusion, a theory that links endometriosis and adenomyosis needs to take into account these differing phenotypes.

### 3.2. Phenotypes of Adenomyosis

Adenomyosis is characterized by the presence of endometrial glands and stroma within the myometrium. This can be localized or diffuse, with different depths, distributions, and densities of myometrial involvement.

It has been proposed that intrinsic (also known as ‘internal’) adenomyosis characterized by increased JZ thickness and tiny mono- or multi-focal cystic structures proximal to the endometrium on magnetic resonance imaging (MRI) and extrinsic (or ‘external’) adenomyosis, which features lesions in the outer portion of the myometrium close to the peritoneal lining, are distinct entities [23,24]. As mentioned, external adenomyosis is often associated with posterior, anterior or lateral DE lesions. This raises the possibility that the external variant may be the product of invasion of the myometrium by DE lesions, but it is also possible that external adenomyosis penetrates through the peritoneum and expands to surrounding structures, causing DE [25].

As summarized in Table 3, using MRI, Kishi et al. [26] distinguished four subtypes of adenomyosis: subtype I (intrinsic) involves lesions directly connected to the eutopic endometrium and is characterized by a thickened JZ; subtype II (extrinsic) refers to cases in which lesions are found in the outer myometrium, and where the JZ seems to be intact in MRI; subtype III (intramural) is the variant in which foci are separated from the JZ and from the serosa; and subtype IV (indeterminate adenomyosis) includes cases that do not conform to any of the above criteria.

Irrespective of origin, different subtypes of adenomyosis seem to have different clinical manifestations, a point initially noted by Kishi et al. [26]. According to Bourdon et al. [27], the clinical profile of women with external adenomyosis differs from that of those with internal adenomyosis. In the former variant, subjects are significantly younger (mean ± SD; 31.9  ±  4.6 vs. 33.8  ±  5.2 years; *p* =  0.006), more often nulligravid (*p* ≤  0.001), and more likely to exhibit an associated endometriosis (*p* <  0.001). Women with internal adenomyosis are more likely to have a history of previous uterine surgery (*p* =  0.002) and heavy menstrual bleeding (HMB) (*p* <  0.001). There were no differences in pain scores (i.e., dysmenorrhea, non-cyclic pelvic pain, and dyspareunia) between the two groups. One important implication is that the different phenotypes may reflect a different pathogenesis.

Recent studies have linked HMB in women with adenomyosis with endometrial fibrosis [28,29,30]. As adenomyotic lesions progress to become more fibrotic, pro-fibrogenic molecules produced by lesions permeate into neighboring JZ and then endometrium, effectively propagating endometrial fibrosis. This increases tissue stiffness or rigidity, which suppresses PGE2 signaling, hypoxia signaling, glycolysis and endometrial HDAC3 expression [28,29,30]. Endometrial downregulation of HDAC3 would, in and of itself, upregulate collagen I, leading to increased endometrial fibrosis [31]. Subsequently, endometrial repair is impaired, resulting in HMB [30]. Thus, it is conceivable that the adenomyotic lesions closer to endometrium, such as internal adenomyosis, are more likely to propagate fibrosis to their neighboring endometrium and, thus, are more likely to impair endometrial repair and cause HMB.

On the other hand, lesions proximal to the serosa, such as external adenomyosis, are unlikely to exert the same effect on the endometrium, and the course could be more protracted. This provides a plausible explanation for the link between superficial adenomyosis and HMB (and likely infertility). Given that the extent of lesional fibrosis correlates positively with the amount of menstrual blood loss in women with adenomyosis who complain of HMB, it is possible that women with internal adenomyosis who do not have HMB have “softer” lesions or fewer adenomyotic foci, with little impact on the endometrium. External adenomyosis is perhaps more closely associated with DE and would be associated more with dysmenorrhea and chronic pelvic pain but less with HMB.

There is an ongoing debate about whether DE represents a variant of adenomyosis or endometriosis. Differences in biomarker expression seem to show a link between DE and extrinsic adenomyosis [25,32,33]. The physical proximity between these two types of lesions also suggests a causal relationship.

In conclusion, just as with endometriosis, adenomyosis has several distinct subtypes or variants with possibly different etiology and clinical manifestations. Type I adenomyosis is associated with the history of iatrogenic uterine procedures and often manifests with HMB and possibly infertility. In contrast, Type II adenomyosis is associated with DE and often manifests with dysmenorrhea and pelvic pain. It is unclear whether Type II is the cause or consequence of DE.

## 4. Epidemiological Evidence of a Link

Although MRI and two- or three-dimensional vaginal ultrasound (2D- or 3D-TVS) represent useful tools for non-invasive diagnosis, as of now, the complete identification of adenomyosis and/or endometriosis remains challenging because invasive tests (laparoscopic or laparotomic cytoreduction or adenomyomectomy) are sometimes needed to confirm the diagnosis and because of the limits of imaging.

The incidence and/or prevalence of the two conditions remains speculative for theoretical [34] and practical reasons [35]. Published literature relies on the diagnosis in series of often highly selected patient subgroups from hospital or clinic statistics or, alternatively, have relied on aggregated population statistics either from reported national data, health insurance, or self-reported symptoms. All of these have significant limitations because they are influenced by the presenting symptoms, availability, access to investigations, and the inability to precisely identify the number of women at risk at any particular time.

### 4.1. Prevalence of Endometriosis

The challenges encountered in quantifying the incidence and prevalence of endometriosis are well recognized [36]. The lack of readily accessible and reliable non-invasive diagnostic tests means that most available estimates are based on subgroups that underwent laparoscopy. Two population studies from the US reported the age-specific incidence among white women. Between 1970 and 1979, the histologically confirmed endometriosis was 160.4/100,000 woman-years in the 15–49 age group with a peak between ages 35 and 44 years [37]. A later study reported an overall incidence of 187/100,000 women-years based on clinical diagnosis, but with a peak at between 25 and 34 years [38]. The incidence of laparoscopically proven endometriosis was higher (298/100,000 woman-years) in the US Nurses’ Health Study II, with a peak of 417/100,000 woman-years in the 25–34 age group [39]. Given that endometriosis is a chronic condition, it is possible, if not probable, that the shift in peak incidence relates to accessibility of care, which may also play a part in the differences in the incidence over the decades. In a nationwide study, the incidence rate of hospital-diagnosed endometriosis in Denmark was 7.89 (95% confidence interval [CI] = 7.80–7.99) per 10,000 woman-years, and the prevalence in 2017 was 1.63%. However, there was a wide regional variation. Women living in northern Jutland had the highest probability of receiving a hospital-based diagnosis of endometriosis (hazard ratio (HR) = 1.13, 95% CI = 1.09–1.18), whereas women living in northern Zealand had the lowest probability (HR = 0.63, 95% CI = 0.60–0.67) compared with eastern Jutland. The authors suggested that the variation may be related to differences in service provision, leading to under diagnosis in northern Zealand [40].

A study from the US involving 332,056 women reported the average incidence of endometriosis to be 24.3 cases per 10,000 woman-years. Of interest, the incidence of endometriosis was similar in women who were surgically or clinically diagnosed and decreased significantly from 2006 to 2015 (*p* < 0.001 for a linear trend over time for each) [41]. 

In a large longitudinal study involving 13,508 women, the cumulative prevalence of clinically confirmed endometriosis was 6.0% (95% CI = 5.8–6.2%) by age 40–44 years, and an additional 5.4% were suspected to have endometriosis. When added together, the cumulative prevalence increased to 11.4% (95% CI = 11.1–11.7%). The study was not able to distinguish endometriosis from adenomyosis, but sensitivity analysis excluding adenomyosis suggested a prevalence of 11% for endometriosis. The peak incidence of endometriosis was in the 30–34 years old group [42].

It has long been recognized that endometriosis can be identified in asymptomatic women. One early study found the incidence to be comparable to that in symptomatic women [43]. A population-wide study from Iceland estimated the crude annual incidence of endometriosis to be 0.1% [44], and a very recent study from the same population confirmed the annual incidence as between 0.1% and 0.15% and the prevalence as 0.6% to 3.6% [45]. Similar incidence and prevalence were reported in other studies [37,46,47,48]. 

Of interest is the study that linked the rate of hospital admission in Sweden because of endometriosis to the country of birth and to socioeconomic variables [47]. The risk of endometriosis may also be linked to the type of contraception used [49].

In conclusion, the incidence of endometriosis in developed countries appears to be in the range of 10 to 40 per 10,000 women-years, and the prevalence ranges from 1.6% to 11%. Data from developing countries are lacking but may be lower due to the higher birth rate.

### 4.2. Prevalence of Adenomyosis

A different picture emerges from investigations of the prevalence of adenomyosis. This also remains uncertain because a number of studies, including recent ones, relied on hysterectomy specimens. Reported incidence of adenomyosis vary widely [50,51,52,53]. The variation may be related to the case selection or to the degree of diligence of histological examination following hysterectomy. Nevertheless, these investigations cannot provide correct information on the prevalence of the condition.

A retrospective, large cohort investigation from the US by Yu et al. [54] involving 333,693 women reported the overall incidence of adenomyosis as 1.03% or 28.9 per 10,000 woman-years. The incidence was highest for women aged 41–45 years and for black compared to white women.

In conclusion, from the data published for the US population, the incidence of adenomyosis seems to be on par with that of endometriosis. With the advance of imaging technology, the reported incidence is likely to be higher since imaging diagnostics can detect lesions in women not undergoing surgery.

### 4.3. Comparative Incidence and Prevalence

A study from one region in Northern Italy reported the incidence of endometriosis in the 15–50 age group as 0.112% and that of adenomyosis as 0.027%. The prevalence of endometriosis was calculated as 1.82% and of adenomyosis as 0.17% [55].

### 4.4. Coexistence of the Two Conditions

Both adenomyosis and endometriosis can coexist, although the frequency of the association varies greatly depending on the study.

Several investigations dealt with the prevalence of endometriosis in women undergoing surgery for adenomyosis. In a study of 710 premenopausal women who underwent hysterectomy due to adenomyosis, 343 (48.3%) had adenomyosis alone, 158 (22.3%) had adenomyosis and endometriosis, 129 (18.2%) had adenomyosis and leiomyomas, and 80 (11.3%) patients had and all three conditions combined [56]. In a study of 291 women with surgically diagnosed adenomyosis, 38.8% were also identified as having endometriosis [57]. The median age of the group was 40 years. Patients with both conditions were older at diagnosis and, at disease onset, had an earlier median age of menarche, a higher rate of menorrhagia, a higher rate of a moderate-to-severe degree of dysmenorrhea. Most (68.1%) cases of endometriosis were ovarian. Only two had DE, but this may be an underestimate of prevalence [57].

In another cohort of 220 patient (47 with adenomyosis and 132 with fibroids) who underwent a hysterectomy for pain (dysmenorrhea, dyspareunia, or non-menstrual pain) and/or menorrhagia and/or a mass effect, endometriosis was identified at operation in 19/47 (40.4%) of those with adenomyosis, in 30/132 (22.7%) of those with fibroids, and in 14/41 (34.1%) of those with both fibroids and adenomyosis [58]. The incidence of endometriosis in cases with adenomyosis reported here is higher than that reported in other studies [50,51], and the difference may be related to different inclusion criteria.

Information is also available on the prevalence of adenomyosis in patients undergoing surgery for endometriosis. In 2005, Leyendecker and Kunz quoted an association in approximately 90% of cases [59]. This prevalence is by far higher than that found in other studies and may be related to a highly selected patient population.

Di Donato et al. [60] retrospectively investigated 1618 women with preoperative clinical and ultrasound diagnosis of endometriosis. They found a 21.8% prevalence of adenomyosis in patients affected by endometriosis and linked adenomyosis to parity, increasing age, intensity of dysmenorrhea, and to the presence of deep infiltrating endometriosis.

It is important to emphasize that a relation between adenomyosis and endometriosis cannot be established through examination of hysterectomy case series when the hysterectomy was not performed because of adenomyosis.

In a study of pregnant women who were diagnosed with endometriosis prior to pregnancy (n = 206), 18.4% had focal and 9.7% had diffuse adenomyosis, as diagnosed by ultrasound. The presence of diffuse but not of focal adenomyosis was associated with delivery of small for gestational age (SGA) infants [61]. In another study involving 1000 women, adenomyosis was identified in 172 (17.2%) of those who underwent surgical treatment for endometriosis. Adenomyosis and fibroids together were present in 11.1% of cases [62].

The critical question when examining the possibility of a pathophysiological link between adenomyosis and endometriosis is whether their coexistence can be a chance occurrence. The literature remains divided on this question [55,57].

Mathematically, the proof that the co-occurrence of both endometriosis (E) and adenomyosis (A) is more than a chance event requires that the probability of having both E and A, i.e., P(EA), be larger than the product of P(E) and P(A), where P(E) and P(A) represent the probability, respectively, of having E and of having A alone. That is, when P(EA) > P(E)P(A), then the co-occurrence is more than a chance event, in contrast to the situation where the conditions are independent. Since P(EA) = P(E)P(A|E), where P(A|E) is the conditional probability of having A in women with E, proof of the link between A and E requires P(E), P(A), and P(A|E) to be known. P(EA) = P(A)P(E|A); hence, P(EA) can also be calculated by the estimates of P(A), P(E), and P(E|A). Since P(E)P(A|E) = P(A)P(E|A) = P(EA) and given that P(E) > P(A), then P(E|A) > P(A|E). This seems to be supported by the published literature, as P(E|A) ranges from 33.6% to 40.4%, whereas P(A|E) is approximately 21.8% [60], assuming that the prevalence of endometriosis and of adenomyosis are more or less the same in different populations. Many clinical studies report P(E|A) or P(A|E), i.e., the proportion of patients with endometriosis among those diagnosed with adenomyosis or vice versa. However, this is insufficient to determine whether P(EA) > P(E)P(A) since P(E) and P(A) cannot be directly calculated from the data provided in these reports. 

In conclusion, whether the often-quoted association between endometriosis and adenomyosis is more than a chance occurrence is still an open question, and its resolution would require more rigorous investigation.

### 4.5. Comparison of Risk Factors

It is also important to consider that adenomyosis and endometriosis have different risk factors. For instance, parity, which is a risk factor for adenomyosis, seems to protect against endometriosis. The age-specific incidence of endometriosis is highest in the age group 31–35, while adenomyosis was highest in the age group 46–50 [55]. 

Factors linked to increased risk of endometriosis include family history, obstructive genital tract anomalies, early menarche, nulliparity, frequent cycles and heavy menstrual bleeding. Women with endometriosis tend to have low body mass indexes and be of Caucasian race or Asian ethnicity. Low birth weight and maternal diethylstilbestrol exposure have also been considered risk factors. In addition, it has been documented that shorter anogenital distance (AGD), a proxy marker for in utero exposure to lower androgen levels, is associated with a higher risk of endometriosis [63,64,65,66] and adenomyosis [67]. However, since the variance of AGD explained by the disease status may be moderate [67], in utero exposure to androgens may not in itself be a major etiological factor.

There may be a link between genetically predicted plasma levels of coagulation factors ADAMS13 (A disintegrin and metalloproteinase with thrombospondin motifs [13]) and vWF (von Willebrand factor) and the risk of endometriosis [68]. While it is recognized that lifestyle factors such as smoking, alcohol, and caffeine intake and exercise can affect steroid metabolism, there are contradictory reports on the effect of alcohol intake [69,70], whilst caffeine and smoking do not seem to increase the risk [71,72].

Adenomyosis, on the other hand, is more common in the 4th and 5th decades. The risk increases with parity, which may be related to the effect on locally produced prolactin, as well as to the need for hysterectomy to confirm adenomyosis. However, the later onset of adenomyosis as compared with the onset of endometriosis may be due to the longer induction period in adenomyosis as compared with endometriosis. In mice, endometriosis can be induced and fully develop in two weeks, while adenomyosis may take 8–10 weeks after EMID.

Adenomyosis has also been linked with a history of depression and the use of antidepressants [73]. There is controversy as to whether the disease is linked to a history of uterine surgery, including dilatation and curettage (D&C), pregnancy termination, or cesarean section [50,74]. Experimental EMID has been demonstrated to cause adenomyosis in mice through mechanisms involving Schwann cell dedifferentiation [75,76].

In conclusion, many risk factors for endometriosis and adenomyosis have been reported. A few of them, such as iatrogenic uterine procedures that have been reported to be associated with increased risk of developing adenomyosis, have been demonstrated experimentally. 

## 5. Common Features

Endometriosis and adenomyosis share as a common feature the presence of ectopic endometrial-like tissue. They are linked to symptoms of heavy periods and dysmenorrhea, but these are common gynecological complaints that can also occur in the absence of identified gross or microscopic pathology. Pain and heavy periods are the symptoms that trigger investigations leading to the identification of endometriosis and adenomyosis, but it is also recognized that the conditions can exist in asymptomatic women. The lack of a simple and cheap non-invasive diagnostic procedure means that the exact incidence remains unknown. On the other hand, there are known differences in that endometriosis is more likely to be linked to infertility whilst adenomyosis has been linked to parity and to procedures involving disruption of the endometrial/myometrial interface.

In conclusion, both endometriotic and adenomyotic lesions undergo cyclic changes that feature bleeding and repeated cycles of tissue injury and repair (ReTIAR) [77]. With this feature, both endometriotic and adenomyotic lesions seem to undergo identical molecular events, such as epithelial–mesenchymal transition (EMT), fibroblast-to-myofibroblast transdifferentiation (FMT), and smooth muscle metaplasia (SMM), and fibrogenesis [77]. These processes may not be causative, but nonetheless shape the tempo and pace of lesion progression.

### 5.1. Correlation between Phenotypes and Clinical Outcome

With regards to adenomyosis, few studies attempted to correlate the various MRI phenotypes to clinical outcomes. A recent systematic review concluded that despite the wide range of objective MRI parameters linked to adenomyosis, their individual diagnostic value remains uncertain [33].

Of importance, JZ characteristics remain the most widely used and investigated parameters with acceptable diagnostic accuracy. At the same time, whereas for years JZ thickness was considered the ‘gold standard’ for diagnosing early stages of adenomyosis, recently, caution has been advocated. Bazot and Darai [24] stated that “further studies are required to determine the performance of direct signs (cystic component) and indirect signs (characteristics of junctional zone)”; Tellum et al. [78] went as far as to conclude that “measuring the JZ thickness is of limited value for diagnosing adenomyosis with MRI and that it should not be used for diagnosing adenomyosis in premenopausal women with moderate disease severity”. Chapron et al. [79] advocated for “an integrated non-invasive diagnostic approach, considering risk factors profile, clinical symptoms, clinical examination and imaging”.

However, emerging data link the extent of lesion fibrosis with the severity of dysmenorrhea in women with endometriosis [80] and with the amount of menstrual blood loss in women with adenomyosis [28,30]. These data seem to suggest that symptom severity increases with disease progress.

### 5.2. The Eutopic Endometrium in Adenomyosis and Endometriosis: Presence and Significance of Gene Mutations

It is recognized that the eutopic endometrium in women with endometriosis or adenomyosis differs from that in unaffected women [81,82]. In a previous review, we concluded that there are similarities as well as differences between the eutopic endometrium in both conditions [11]. More recently, mRNA global expression profiling (mRNA-Seq) identified 72 differentially expressed (66 upregulated and 6 downregulated) genes in secretory samples of the eutopic endometrium of women with endometriosis [83].

Herndon et al. [84] identified 140 up-regulated and 884 down-regulated genes in the eutopic endometrium in adenomyosis compared to unaffected controls in the proliferative phase. However, there was no overlap when the 50 top differentially regulated genes in the eutopic endometrial biopsy in women with adenomyosis versus control from this study were compared to the differentially expressed genes in endometriosis in the study by Zhao et al. [83].

Bulun et al. [85] used deep-sequencing analyses of epithelial cells and of adjacent basalis endometrial glands in adenomyosis to identify recurring *KRAS* mutations in both cell types. This research suggests that adenomyosis originates from basalis endometrium. Epithelial cells of eutopic endometrium in adenomyosis and co-occurring endometriosis share identical KRAS mutations, which points to both conditions featuring oligoclonal tissues arising from endometrial cell populations carrying a specific driver mutation that most commonly affects the *KRAS* gene. Xiang et al. [86] compared the eutopic endometrium in the proliferative phase of women with and without adenomyosis and reported 258 up-regulated and 115 down-regulated genes. Of those genes up regulated in the eutopic endometrium in endometriosis, 14 were also differentially upregulated in eutopic endometrium in adenomyosis, although none of the down-regulated genes were amongst those downregulated in adenomyosis.

Gan et al. [87], using a sequencing technique coined mRNA-Seq, compared the expression of genes in endometrial stromal cells obtained during the secretory phase from women with adenomyosis to those from controls. There was a total of 458 up-regulated and 363 down-regulated genes in adenomyosis. However, in the study by Zhao et al. [83], none of the top 10 up- or down-regulated genes in stromal cells in adenomyosis were also differentially expressed in the eutopic endometrium in endometriosis.

There are indications of *KRAS* mutation as a key factor in a shared pathophysiology of endometriosis and adenomyosis. It has long been demonstrated that structural continuity exists between adenomyotic glands and the eutopic endometrium. Therefore, it is not surprising that both exhibit the same mutations, as demonstrated in the *KRAS* gene [85]. Identical mutations were identified also in the superficial peritoneal, deep-infiltrating, and ovarian endometriosis [88].

In a landmark investigation by Inoue et al. [14], identical mutations of *KRAS* were found in patients with coexisting endometriosis and adenomyosis. Shared *KRAS* and non-*KRAS* mutations were found in some cases with co-occurring adenomyosis and endometriosis (Figure 1). However, the study also showed evidence of clonal heterogeneity in that adenomyosis, and endometriosis may contain different *KRAS* mutations and there can be mutations in disease free individuals. Since endometrium from apparently healthy women could also harbor *KRAS* mutations, which confer proliferative advantage without causing any overt pathology [89], we have thus argued that all endometrial aberrations reported so far are neither sufficient nor necessary to cause endometriosis or adenomyosis [1].

In conclusion, it has now been proven that both endometriotic and adenomyotic lesions contain cancer-associated mutations. What is still debatable is the precise roles of these mutations and whether, by conferring selective advantage, they drive the development of ectopic endometrium or whether they are merely part of the fibrogenesis process [90]. Sequencing data unveiled enormous heterogeneities in cellular populations that were previously thought to be one singular cell type. However, this data represents only a single snapshot of the lesional progression. Much more research is needed to expand these findings and build a clearer global picture of lesional development and its containment. 

### 5.3. Molecular Aberrations in Adenomyosis and Endometriosis

Many studies have reported differences between the eutopic endometrium in women with adenomyosis or endometriosis compared to unaffected controls. These differences have been linked to increased adhesion, invasion, and survival and also to the enhanced ability to evade immune mechanisms and to stimulate neo-vasculogenesis at ectopic sites. On the other hand, there is evidence that the differences identified in the eutopic endometrium can be induced by the presence of ectopic endometrium. Studies that compared the eutopic endometrium from women with adenomyosis and with endometriosis also reported molecular differences between the two groups [13], including immune and proliferation markers, HLA-DR, integrins (VLA-2, 3, 4, 5, 6), E-cadherin, VLA 2–4 and E-Cadherin, eNOS, Bcl-2, intra-epithelial leukocytes CD45+, CD43+, oxidative stress, free radicals, superoxide dismutase, glutathione peroxidase (GPx), cytokines and inflammatory mediators, COX-2 expression, xanthine oxidase (XO), catalase, PGP 9.5 and NF protein, and aromatase. This suggests that predisposing factors may be different between the two conditions.

### 5.4. Effect of Treatment

An intriguing question is whether the effect of treatment can provide any insight about pathophysiology or a link between endometriosis and adenomyosis. Both conditions are well recognized to be responsive to the administration or the withdrawal of steroids. However, there is no effective ‘cure’ that can lead to the eradication of endometrium from ectopic sites, whether within or outside the uterus. It has historically been recognized that symptoms attributed to endometriosis and/or to adenomyosis cease with the onset of menopause. However, the effect can be reactivated following exogenous steroid administration. On the other hand, the effects of hormone manipulation other than total steroid withdrawal are not uniform and little is understood about the causes or predictors of treatment failure.

## 6. Conclusions

Significant challenges remain before the hypothesis can be accepted that adenomyosis and endometriosis are a single disease with unified pathophysiology and, indeed, adenomyosis is not simply endometriosis of the uterus. 

There are wide differences in estimates of the coexistence of both conditions. The higher coincidence has been reported in small studies of highly selected groups. Obtaining population-wide data is hampered by methodological and practical constraints. There is some overlap in clinical features, but symptoms such as pain and abnormal menstruation can occur in the absence of identifiable disease. Uterine affection with adenomyosis varies in its extent and depth, and endometriosis varies widely in its extent and the exact organ(s) affected. These differences pose the question of whether subtypes of endometriosis and adenomyosis have different etiopathologies, and, in turn, this challenges whether all share a common etiology. Existing data, such as EMID-induced adenomyosis, seem to suggest that, at least for adenomyosis, there can be more than one cause. Perhaps the most significant development is the finding of common mutations, particularly of *KRAS*, in adenomyosis and endometriosis. Whilst this can lend support to the notion of shared predisposing factors, it is insufficient evidence of shared causation. In fact, *KRAS* mutation is also reported in the endometrium of women who are apparently healthy, free of any pathology.

## Figures and Tables

**Figure 1 biomolecules-14-00032-f001:**
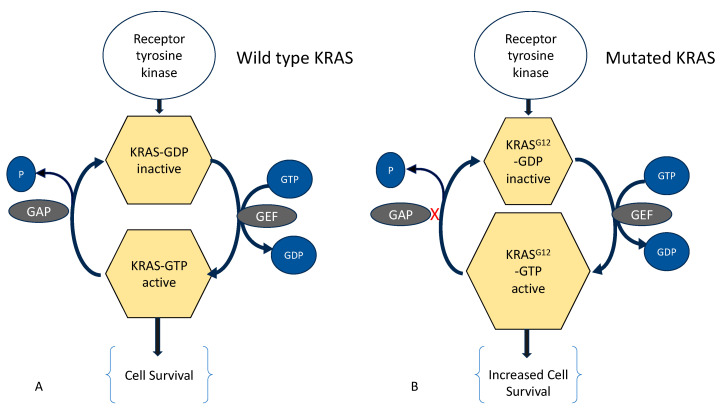
(**A**) Wild type KRAS can hydrolyze a bound guanosine triphosphate (GTP) into guanosine diphosphate (GDP), but it requires binding of GTPase-activating proteins (GAP), which accelerates hydrolysis of KRAS-GTP to the inactive KRAS-GDP. (**B**) G12 mutation of KRAS renders it insensitive to GAP, leading to accumulation KRAS-GTP. In turn, this leads to enhanced activation of—amongst others—cell survival pathway. GEF: Guanine Nucleotide Exchange Factor, (X) indicates lack of binding, (P) phosphate.

**Table 1 biomolecules-14-00032-t001:** Search items and retrieved articles.

	Database: Ovid MEDLINE(R) ALL <1946 to 13 October 2023>	
	Search Strategy:	No. of Articles
1	adenomyosis.mp. or Adenomyosis	3775
2	endometriosis.mp. or Endometriosis	33,643
3	etiology.mp.	2,939,921
4	pathophysiology.mp.	179,252
5	3 or 4	3,080,247
6	1 and 5	877
7	2 and 5	8862
8	7 and 8	496
9	from 8 keep 2, 12, 26, 27, 34, 77, 122, 151, 166, 227, 233, 320, 324, 430	14
10	Prevalence/or prevalence.mp.	896,010
11	incidence.mp. or Incidence	1,039,284
12	10 or 11	1,830,071
13	1 and 12	436
14	2 and 12	2663
15	13 and 14	247
16	1 and 2	1972
17	12 and 16	247
18	from 17 keep 17, 38, 51, 62, 66, 78, 79, 85, 86, 91, 101, 102, 104, 106, 113, 125, 145, 147, 149, 162, 166, 204	22

**Table 2 biomolecules-14-00032-t002:** Reported sites of extra-pelvic endometriosis.

Affection of the Abdominal Wall	Primary Abdominal Wall Endometriosis (No Previous Scar)
	Scar endometriosis
	Umbilical endometriosis
	Inguinal endometriosis
Perineal endometriosis	Primary (no previous surgery)
	Following scarring
Visceral endometriosis	Kidney
	Liver
	Pancreas
	Biliary tract
	Diaphragm
Thoracic endometriosis	Diaphragm
	Pleura pneumothorax
	Lung parenchyma
Nose	
Central nervous system	Brain
	Lumbar vertebrae
	Conus medullaris
Extra-pelvic muscles	
Peripheral nerves	

**Table 3 biomolecules-14-00032-t003:** MRI Classification of Adenomyosis according to Kishi et al. [26].

Subtype	Description
I	Adenomyosis that occurs in the uterine inner layer without affecting the outer structures.
II	Adenomyosis that occurs in the uterine outer layer without affecting the inner structure
III	Adenomyosis that occurs solitarily without relationship to structural components
IV	Adenomyosis that did not satisfy the above criteria

## Data Availability

Not applicable.

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
