# Peer review of "Are Adenomyosis and Endometriosis Phenotypes of the Same Disease Process?"

_biomolecules, 2023, doi:10.3390/biom14010032_

Round 1

Reviewer 1 Report

Comments and Suggestions for Authors

In the manuscript “Are Adenomyosis and Endometriosis Phenotypes of the Same Disease Process?” submitted to Biomolecules, the authors describe that adenomyosis and endometriosis represent different phenotypes of a single disease. On the other hand, a common etiopathology for adenomyosis and endometriosis has been suggested because both conditions entail the presence of endometrial tissue at locations other than the uterine endometrium. Finally, the authors suggest that significant challenges remain before the hypothesis could be accepted that adenomyosis and endometriosis are a single disease with unified pathophysiology. Furthermore, the authors mentioned common mutations of KRAS in adenomyosis and endometriosis, but there is insufficient evidence of shared causation.

The manuscript is very interesting. The authors have governed many reports and discussed them in a coherent, simple, and straightforward.

However, this review manuscript describes mainly the differences and similarities of pathological characteristics between adenomyosis and endometriosis, not molecular mechanisms. The authors should focus on the differences and similarities of molecular mechanisms in adenomyosis and endometriosis for publication in the Biomolecules journal. Furthermore, the authors should include illustrations to be easily understood by readers.

Comments on the Quality of English Language

This is a very well-written.

Author Response

REVIEWER 1

In the manuscript “Are Adenomyosis and Endometriosis Phenotypes of the Same Disease Process?” submitted to Biomolecules, the authors describe that adenomyosis and endometriosis represent different phenotypes of a single disease. On the other hand, a common etiopathology for adenomyosis and endometriosis has been suggested because both conditions entail the presence of endometrial tissue at locations other than the uterine endometrium. Finally, the authors suggest that significant challenges remain before the hypothesis could be accepted that adenomyosis and endometriosis are a single disease with unified pathophysiology. Furthermore, the authors mentioned common mutations of KRAS in adenomyosis and endometriosis, but there is insufficient evidence of shared causation.

The manuscript is very interesting. The authors have governed many reports and discussed them in a coherent, simple, and straightforward.

Reply: Many thanks

However, this review manuscript describes mainly the differences and similarities of pathological characteristics between adenomyosis and endometriosis, not molecular mechanisms. The authors should focus on the differences and similarities of molecular mechanisms in adenomyosis and endometriosis for publication in the Biomolecules journal.

Reply: Comments on molecular mechanisms have been made throughout the manuscript whenever relevant. At any rate, we have added a new section (5.3. Molecular aberrations in adenomyosis and endometriosis) to strengthen this aspect.

Furthermore, the authors should include illustrations to be easily understood by readers.

Reply: We have added three Tables and two Figures to the revised text.

Comments on the Quality of English Language

This is a very well-written.

Reply: Many thanks

Reviewer 2 Report

Comments and Suggestions for Authors

In manuscript:” Are Adenomyosis and Endometriosis Phenotypes of the Same Disease Process?” by Habiba et al., authors made a review about the theory that adenomyosis and endometriosis represent different phenotypes of a single disease. Authors conclude that adenomyosis is not simply endometriosis of the uterus and the hypothesis that adenomyosis and endometriosis represent different phenotypes of single disease, can’t be confirmed.

General comments: The topic of this manuscript could be interesting for readers because to date, there is no clear answer whether adenomyosis is endometriosis of the uterus and whether these two diseases share common molecular alterations and are phenotypically similar. The manuscript is very well written; topic is presented correctly, and it is easy to understand and follow. Still, despite of interesting topic, several questions raised that should be clarified.

1. In Materials and methods section please describe more precisely how the literature review was done, how exactly the inclusion/exclusion criteria were used (perhaps flow chart) illustrating the identification and screening of literature (how many duplicates were identified (with numbers) and removed, papers in other language than English, animal studies etc.

2. The authors also describe the molecular changes related to endometriosis and adenomyosis (5.2. The Eutopic Endometrium in Adenomyosis and Endometriosis). The title of this section should indicate the content of the section. Also, please indicate clearly where are gene-expression studies and where are studies involving DNA mutations. Currently this section is a mixture of both. Also, there are many molecular studies (including gene-expression, single cell studies both in endometriosis and adenomyosis) that are not discussed in this section. As the topic of this paper is about shared phenotypes, perhaps it would be reasonable not to discuss the molecular findings that are extensively discussed earlier in detail by Bulun et al.2023. If the authors would like to keep this section, please provide more extensive overview about the molecular similarities and differences between adenomyosis and endometriosis.

 3.      I would like to see summary sentences at the end of larger sections whether authors accentuate their opinion about the topic and summarize the current knowledge in context of this paper topic.

Author Response

REVIEWER 2

In manuscript “Are Adenomyosis and Endometriosis Phenotypes of the Same Disease Process?” by Habiba et al., authors made a review about the theory that adenomyosis and endometriosis represent different phenotypes of a single disease. Authors conclude that adenomyosis is not simply endometriosis of the uterus and the hypothesis that adenomyosis and endometriosis represent different phenotypes of single disease, can’t be confirmed. 

Reply: Thanks. This is exactly our conclusion. We tried to make this clear at the end of our review (Item 6)

General comments: The topic of this manuscript could be interesting for readers because to date, there is no clear answer whether adenomyosis is endometriosis of the uterus and whether these two diseases share common molecular alterations and are phenotypically similar. The manuscript is very well written; topic is presented correctly, and it is easy to understand and follow. 

Reply: Many thanks

Still, despite of interesting topic, several questions raised that should be clarified. In Materials and methods section please describe more precisely how the literature review was done, how exactly the inclusion/exclusion criteria were used (perhaps flow chart) illustrating the identification and screening of literature (how many duplicates were identified (with numbers) and removed, papers in other language than English, animal studies etc.

Reply: The reviewer is correct. We have added all the details we have.

The authors also describe the molecular changes related to endometriosis and adenomyosis (5.2. The Eutopic Endometrium in Adenomyosis and Endometriosis). The title of this section should indicate the content of the section.

Reply: In compliance of the Reviewer’s request, we have modified the title as follows: “The Eutopic Endometrium in Adenomyosis and Endometriosis: presence and significance of molecular and gene mutations”. We have also added a new section “5.3. Molecular aberrations in adenomyosis and endometriosis”.

Also, please indicate clearly where are gene-expression studies and where are studies involving DNA mutations. Currently this section is a mixture of both. Also, there are many molecular studies (including gene-expression, single cell studies both in endometriosis and adenomyosis) that are not discussed in this section. As the topic of this paper is about shared phenotypes, perhaps it would be reasonable not to discuss the molecular findings that are extensively discussed earlier in detail by Bulun et al.2023. If the authors would like to keep this section, please provide more extensive overview about the molecular similarities and differences between adenomyosis and endometriosis.

Reply: We agree that the recent article by Bulun et al provided very valuable information on the topic of “gene mutations”. At the same time, we would like to keep the section for completeness and to give credit to other investigators.

I would like to see summary sentences at the end of larger sections whether authors accentuate their opinion about the topic and summarize the current knowledge in context of this paper topic.

Reply: We have added “Conclusions” at the end of post sections.

Reviewer 3 Report

Comments and Suggestions for Authors

Dear authors, 

in the following you find some comments on the manuscript you submitted to Biomolecules: 

- Line 199: you are mentioning MRI and 3D-TVS. I suggest that you use the term TVS or you add 2D-TVS as this technique represents the most frequent approach in diagnosing adenomyosis and deep endometriosis. A variety of recent papers could be cited here and I propose to mention sensitivity and specificity of imaging techniques.

- Line 201: what do you mean by "invasive tests". Please specify and explain.

Comments on the Quality of English Language

- minor spelling improvements required

Author Response

Reviewer 3

Dear authors, 

in the following you find some comments on the manuscript you submitted to Biomolecules: 

- Line 199: you are mentioning MRI and 3D-TVS. I suggest that you use the term TVS or you add 2D-TVS as this technique represents the most frequent approach in diagnosing adenomyosis and deep endometriosis. A variety of recent papers could be cited here and I propose to mention sensitivity and specificity of imaging techniques.

Reply: We would be happy to add the specifications requested by the reviewer, but first wish to stress that this issue is totally outside the purpose of our review. Furthermore, whereas for adenomyosis TVS may be a valid diagnostic tool, its usefulness in the major variants of endometriosis is still debated; this is especially true for superficial peritoneal disease.

For these reasons, before modifying the text we would like to receive guidance from the Editors.

- Line 201: what do you mean by "invasive tests". Please specify and explain.

Reply: We have specified that this expression is synonymous of “surgical diagnosis”.

Comments on the Quality of English Language

- minor spelling improvements required

Reply: We have done our best to correct these minor inaccuracies.

Round 2

Reviewer 1 Report

Comments and Suggestions for Authors

The authors responded to my comments well.